# Determining the Anti-Erosion Efficiency of Forest Stands Installed on Degraded Land

Mircea Moldovan [1] , Ioan Tăut [1,2,*], Florin Alexandru Rebrean [2,*], Bartha Szilard [3], Iulia Diana Arion [2] and Marcel Dîrja [2]

[1]   National Institute for Research and Development in Forestry (INCDS) "Marin Drăcea", Cluj Branch, Street Horea, no. 65, 400202 Cluj Napoca, Romania
[2]   Department of Forestry, University of Agricultural Sciences and Veterinary Medicine Cluj-Napoca, Street Calea Mănăștur, no. 3-5, 400372 Cluj Napoca, Romania
[3]   Department of Forestry and Forest Engineering, University of Oradea, Gen. Magheru Street, no. 26, 410048 Oradea, Romania
*   Correspondence: ioan.taut@usamvcluj.ro (I.T.); florin-alexandru.rebrean@usamvcluj.ro (F.A.R.)

**Abstract:** Erosion caused by human activities is one of the reasons for forest soil degradation worldwide, with a direct impact on forest stands development, including reduced forest productivity. Therefore, in order to establish sustainable stand management practices, it is essential to assess soil losses in various forestry activities. Moreover, this phenomenon is studied little in stands, especially those established on degraded land. In Romanian geographical conditions, where sloping land is up to 67% of the territory and is influenced by natural factors as well as intense human activities, the soil and vegetation suffer serious ecological imbalances. In order to achieve the proposed objectives regarding the evaluation of stands in terms of anti-erosion effectiveness, we analyzed the consistency and the number of trees on the surface, the weight of the seedlings, and the surface runoff from the perspectives of rainfall and soil retention. In the two stands included in this study, the influence of rain intensity was 39% in compartment 49 and 38% in compartment 73, directly influencing surface runoff. The ground retention's influence on surface runoff was 28% in both compartments. The indirect surface runoff was influenced by the consistency of the stands and by the degree of proximity of the crowns, which directly influenced the intensity of rain. In addition to analyzing these two parameters (rain intensity and ground retention), it was also observed that the degree of proximity to the crowns directly influenced the intensity of the rainfall within the forest, which, in turn, indirectly affected the runoff.

**Keywords:** rain intensity; runoff; runoff plots; stands consistency

## 1. Introduction

The natural phenomenon of erosion has been active throughout geological time until now, shaping the surface of the land. Today, the phenomenon of erosion causes significant effects on the natural environment, which is exacerbated by human activities [1]. Among human activities, those that most strongly influence the processes of erosion are transportation and sedimentation, deforestation, and the cutting and burning of vegetation. Under certain circumstances, the erosion rate can be from 100 to 1000 times higher than the geological erosion generated by natural factors, amounting to approximately 25 tons/km$^2$/year [2].

Ibrahim [3] states that the rate of soil erosion is often higher than the rate of soil formation. Erosion, being a natural phenomenon, is in balance with soil formation (<1.0 tons/hectare) in the plain areas and slightly higher in the mountain areas. Harmful effects of human activities disturb the balance, resulting in a higher rate of erosion compared to the rate of soil formation. Erosion caused by human activities is one of the causes of forest soil degradation worldwide, with a direct impact on stand development, including reduced forest productivity [4–7]. Therefore, in order to establish sustainable stand

management practices, it is essential to assess soil losses in various forestry activities [8,9]. The main causes of erosion generated by human activities are inadequate or intensive management practices, land-use over-exploitation, deforestation, and overgrazing, which all are amplified by climatic and geomorphological factors [10]. Aburto et al. [9] discovered that about 40% of degraded lands worldwide confront accelerated erosive processes, and it is one of the main degradation processes threatening soil resources worldwide.

After years of research, the problem of soil erosion still persists despite the fact that, in most cases, there are adequate technical solutions, such as the use of soil-friendly agricultural practices (using equipment with low impact on soil compaction, the use of the latest generation of chemical pesticides or even biological ones, reducing pollution, terraced farming, alternating deep-rooted and shallow-rooted crops, etc.). This fact raises the question of why soil conservation is not implemented faster. Studies show that the implementation of soil conservation measures depends on a multitude of factors, but it is also clear that rapid change in agricultural systems occurs only when the farmer has a clear economic incentive [11,12]. The rehabilitation of degraded areas and the protection of biodiversity, ecosystem services, and human well-being is achieved through specific ecological reconstruction works. Few studies have translated ecological theory into real reconstruction practices that can be easily used by different stakeholders [13]. However, the use of functional characteristics for planning improvement strategies has been suggested, as they are the main environmental qualities underlying process and service ecosystems [6].

Forest cultures with pioneer species, such as pines, have been considered to be effective in controlling soil erosion [14], and, as a result, extensive afforestation programs were promoted worldwide. Many studies have presented the positive aspects of plantations in reducing soil erosion in previously deforested lands [9,15]. In areas where the hydraulic erosion process is present, forests reduce soil erosion in the following ways: tree crowns reduce rainfall kinetic energy and sediment concentration, surface litter and shrubs absorb rainfall and increase soil infiltration by providing organic matter, and roots can consolidate and hold soil [16].

A number of studies that focus on the runoff associated with the retention of ground litter, shrubs, and flora, have been conducted in the past. Li et al. [17] investigated the relationship between ground litter and surface runoff in northern China. Prosdocimi et al. [18] performed an experiment in Mediterranean vineyards which involved determining the erosion caused by surface runoff. In Japan, Miyata et al. [19] studied surface runoff generation and soil erosion in mature Japanese cypress plantations. To date, there has been little investigation into the role of ground litter, shrubs, and flora on surface runoff dynamics that are naturally or artificially regenerated. One example is a study by Gomyo & Kuraji [20], in which they demonstrated the effects of litter removal on the catchment scale. The study showed that 3-year runoff increased by 2.7% (80.3 mm) post-treatment. In addition, peak runoff was up to 1.5 times higher. These results suggest that the effects of litter interception were greater than those of surface runoff without litter [21,22].

The interception of surface runoff by canopies, litter, and shrubs has been studied very little because it involves lengthy time observation, labor, and high-performance equipment. Worldwide, especially in Japan, studies focusing on the hydrological role of the canopy litter and shrub forests are scarcer because most efforts have focused on cypress plantation forests. Most studies that were conducted in Japan were related to the effects of thinning of conifer plantations, whereby the thinning reduces canopy interception and may increase the likelihood of floods [23,24].

In Romanian natural environments, where the sloping land is up to 67% of the territory and is influenced by natural factors, as well as the intense human activity from the end of the 19th century and the beginning of the 20th century, the soils and vegetation suffered serious ecological imbalances, such as massive soil degradation and pollution. For this reason, the erosion of land and the increase in the frequency of torrential processes (temporary streams that appear as a result of torrential rains or sudden melting of snow, resulting in erosion phenomena) lead to the emergence of semi-arid areas [25]. Dîrja [26] states that once started,

the phenomenon of erosion is accelerated over time until its combating becomes difficult, especially from technical and economic points of view. Therefore, measures must be taken to prevent and combat erosion at early stages when the necessary work required is minimal and involves little cost. In Romania, ecological reconstruction through the afforestation of degraded lands started after 1948 using pines, especially black and Scots pines [27]. These species have been widely used due to low requirements for pedological and climatic factors and can grow on lands with strong erosion [28].

The main objectives of the afforestation of degraded lands were the valorization of these lands both ecologically, by creating new stands, and economically, by using the resulting loamy mass. Conifer species outside the natural area have fast growth, with short-cycle production periods (about 50 years) and were intended for the pulp industry [29]. In Romania, the afforestation of degraded lands gained momentum after 1948, with the main species represented being pines, especially black pine and Scots pine. The most important areas where afforestation works were carried out were the Bistriței Valley, the Vrancea area, the Apuseni Mountains, the Transylvanian Plain, the South of Moldova, the Argeș basin, the banks of the Danube in the Drobeta Turnu Severin area, and Moldova Nouă [27]. The strategies regarding afforestation on degraded lands from that time were, on the one hand, the improvement and introduction of these lands into the forestry circuit and, on the other hand, the creation of forests with resinous species outside the natural range that were fast-growing, with short production cycles (about 50 years). These stands had a secondary destination, meaning for the pulpy industry, but since 1990, these forests have been assigned a protective role, being managed according to functional group I, and their life span being until they can no longer fulfill their protective functions [29].

The anti-erosion effect of stands established on degraded land has been studied little. Therefore, in this study, the behavior of some pine stands installed on lands with erosion phenomena was followed. The anti-erosion efficiency was quantified by determining the surface runoff through the influence of the intensity of rain and of the soil's retention of it. The results obtained can be used to improve current policies regarding the management of existing stands by establishing new stands on degraded lands.

## 2. Materials and Methods

### 2.1. Location

The amelioration perimeters where the study took place can be found in the area of Diviciorii Mari (Figure 1), administered by Forest District Gherla. The main characteristics are presented in Table 1.

**Table 1.** The main characteristics of the studied perimeters.

| Compartment | Subcopartment | Coordinates | Soil | Orientation | Altitude (m) | | | Slope (°) | Stand Age (years) |
| --- | --- | --- | --- | --- | --- | --- | --- | --- | --- |
| | | | | | **Min** | **Max** | **Med** | | |
| 49 | 49A | 46°59′6.73″ N24°4′28.96″ E | Molic Regosol | South-west | 312 | 381 | 347 | 38 | 50 |
| | 49B | 46°59′11.46″ N24°4′31.85″ E | Molic Regosol | North-west | 318 | 395 | 357 | 40 | 40 |
| 73 | 73 | 46°59′25.08″ N24°3′21.76″ E | Typical Regosol | South-west | 311 | 431 | 371 | 37 | 30 |

In order to assess the behavior of the stands in terms of anti-erosion effectiveness, the consistency of the forests was determined by measuring crown projection, the tree number per area, and the percentage of seedlings, which were mapped in six areas in sub-compartment 49A, four in 49B and six in compartment 73. The consistency of the stands, in association with shrubs, flora and litter, prevents surface runoff and thus, soil erosion is reduced by their ability to capture some of the rainfall and reduce the rate of fall, thus reducing the intensity of rain in the stands [30–32].

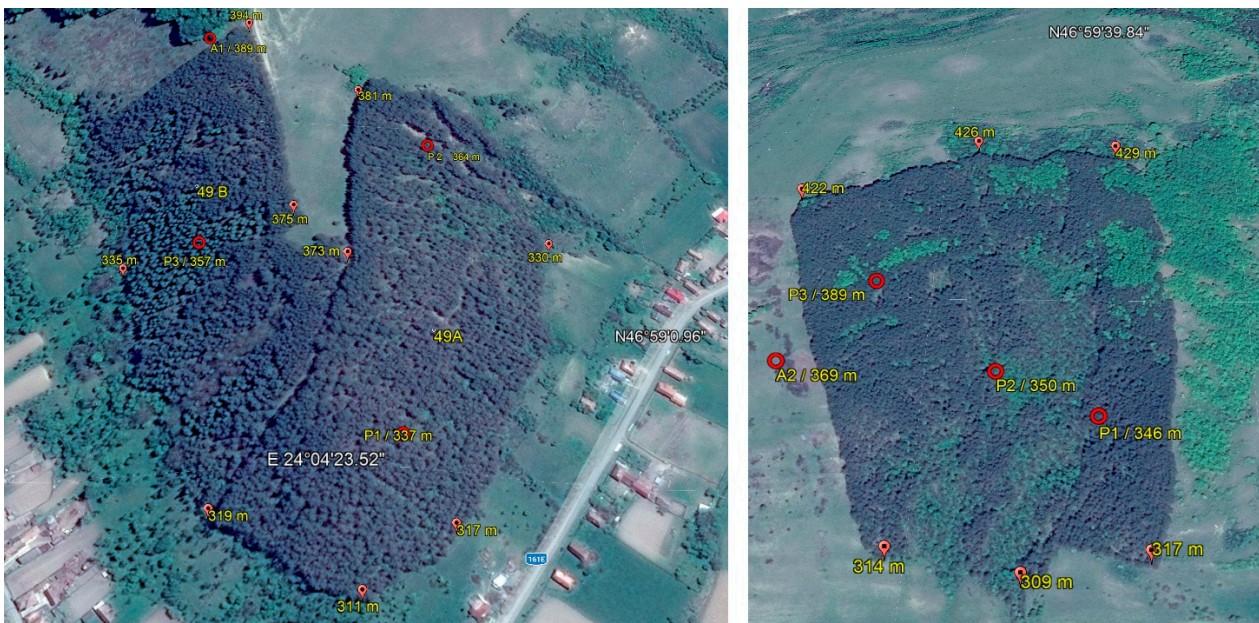

**Figure 1.** Location of runoff plots (Source: Google Earth).

Experimental surfaces were placed using the Google Earth tool, where the optimal areas for the location of the runoff plots were located within the studied stands. Attempts have also been made to use satellite imagery to determine erosion, but the results have been inconclusive: the main cause being the crowns of the trees, which cover the ground.

The surfaces were placed according to the grid method, statistically covering each surface, having a circular shape and an area of 200 m$^2$.

## 2.2. Methodology

The determination of surface runoff was made in six runoff plots located within the stands and two located near them in grassy land [33,34], which involved measuring the depth of runoff (Figure 2). They had a rectangular shape, with an area of 200 m$^2$.

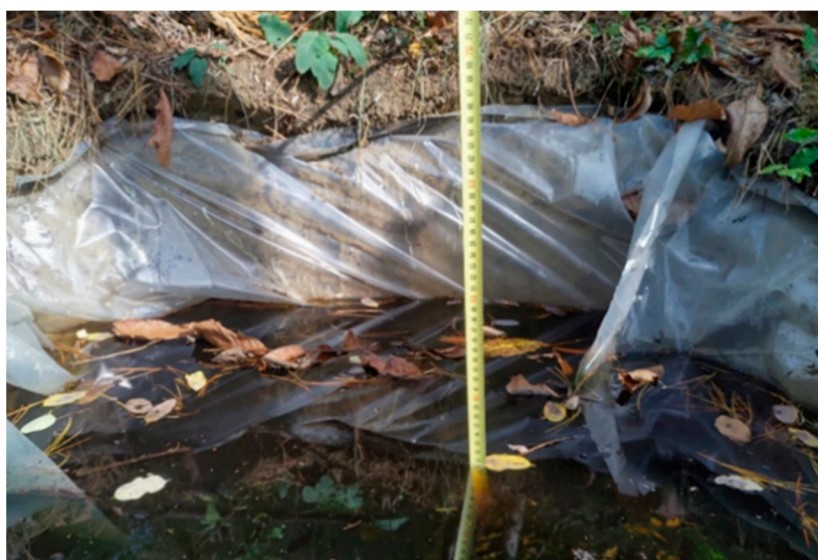

**Figure 2.** Measuring the depth of runoff.

Simultaneously with the measurement of the runoff carried out in the field, measurements were made both inside and outside the studied stands with the help of rain gauges.

Linear regression was used to quantify the influence of rain intensity and soil retention on runoff, calculated using the Microsoft Excel Data Analysis function. To ensure the accurate determination of the amounts and intensities of rain during the research period (16 June 2019–20 July 2021), the periods of precipitation were measured in minutes, and for the rain intensity, the following formula was applied:

$$i = P/T \tag{1}$$

where P = volume of precipitation (L/SQM) and T = rain duration (minutes) [35].

The other influencing factor on surface runoff is represented by soil retention, which is indirectly influenced by the development of stands. It is defined by the amount of water lost through retention in the litter and infiltration in the soil and is analytically quantified by the formula

$$S = P - Z - I \tag{2}$$

where S = surface runoff, P = amount of precipitation, Z = retention in the litter, and I = soil infiltration, reported to 200 square meters [26]. These parameters were used because the anti-erosion effect of forests can be quantified indirectly. The retention at the ground level was determined by the difference between the theoretical drainage (P) (quantity of water drained per surface unit without disturbances caused by retention at the soil level and infiltration into the soil, from the total amount of rainfall measured with the rain gauge) and the average surface runoff (S) measured on the basin (Figure 2).

In order to establish a connection between the intensity of the rain and the surface runoff, meaning between soil retention and surface runoff, the simple linear regression was used with the formula:

$$y = a + bx \tag{3}$$

where a is the incept or free term, a constant that represents the height at which the line intersects the Y axis, and b is the coefficient of regression (slope of the line) and represents the value by which y changes when x increases by one unit [36]. According to the specialized literature, the degrees of freedom used in the experiments are 14, and the theoretical "t" are: t 0.05 = 2.145, t 0.01 = 2.977, and t 0.001 = 4.140 [36].

## 3. Results

The protective actions of tree crowns, shrubs, flora, and litter could not be directly correlated with erosion. In this case, there were necessary data on the quantities of water from precipitation reaching the ground (Figure 3) and the rainfall intensity (Figure 4), which are directly influenced by the phytosanitary condition and, thus, the consistency of the forests.

Within plot 49, tree cover is 80%, the average number of trees per 200 square meters is 15.6, and the number per hectare is 780 trees. Within plot 73, the coverage of the forest is 70%, the average number of trees per 200 square meters is 21.11, and the number per hectare is 1055 trees (Figure 5).

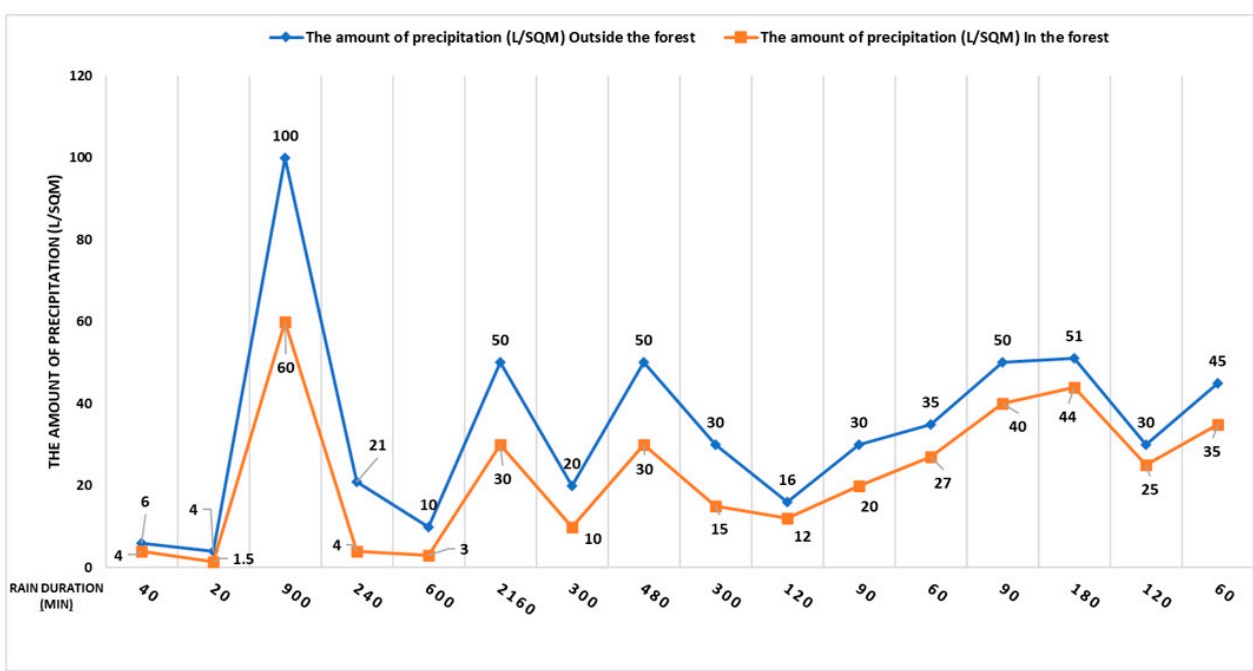

**Figure 3.** Volume of precipitation reaching the ground inside and outside the stands, expressed in L/SQM between 16 June 2019–5 July 2020.

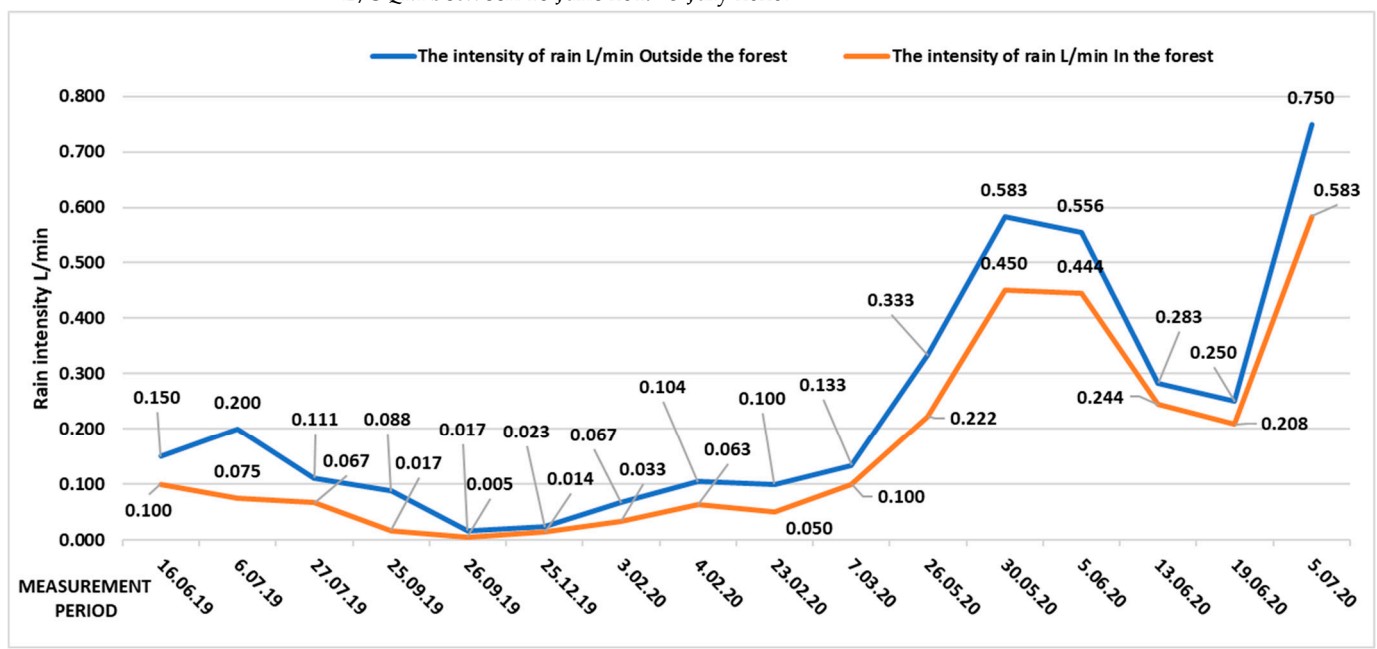

**Figure 4.** Rainfall intensity inside and outside the stands, expressed in L/MIN between 16 June 2019–5 July 2020.

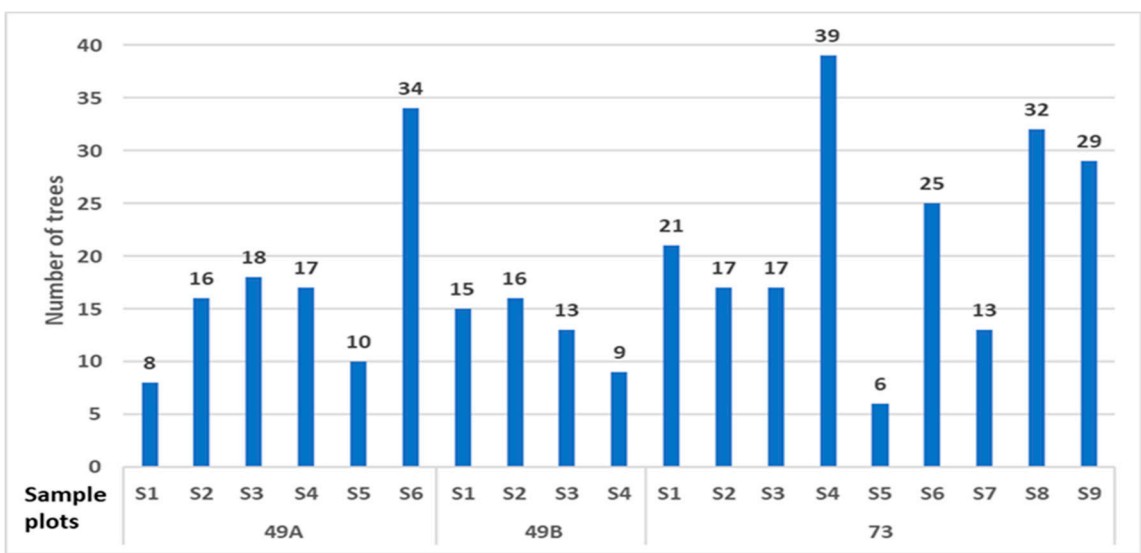

**Figure 5.** Results of the inventory in the improvement perimeters.

In addition, the proportional coverage of natural seedlings was determined. Thus, in sub-compartment 49A, the component species of the seedlings are about 20% false acacia, 30% walnut, and 40% sessile oak. In sub-compartment 49B, the component species are about 1% false acacia, 5% walnut, 15% sessile oak, and 25% beech. In compartment 73, the seedlings are composed of about 30% ash, 40% oak, 40% fluffy oak, and 50% Hungarian oak (Figure 6).

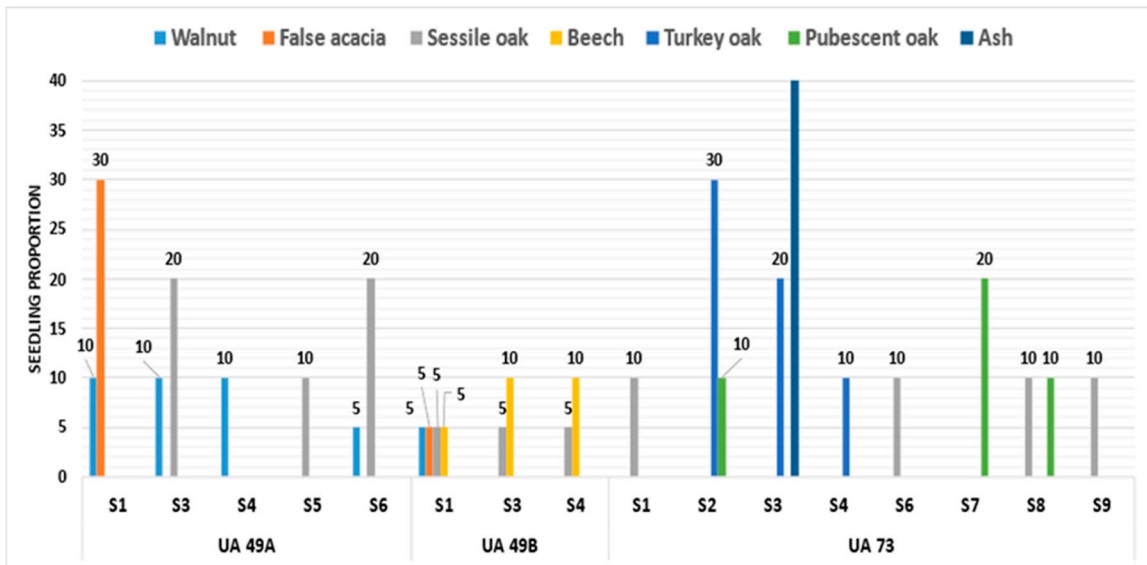

**Figure 6.** Component species and percent of seedlings participation in each experimental plot.

Within compartment 49, the regression equation obtained is: y = 206.73 + 448.19x (r-square = 0.39), and its graphic presentation is in Figure 7. According to this, the surface runoff is influenced by the intensity of the rain to the extent of 39%. Comparing the calculated "t" value 3.282 with the theoretical t 0.01 = 2.977, it follows that the obtained results are significant.

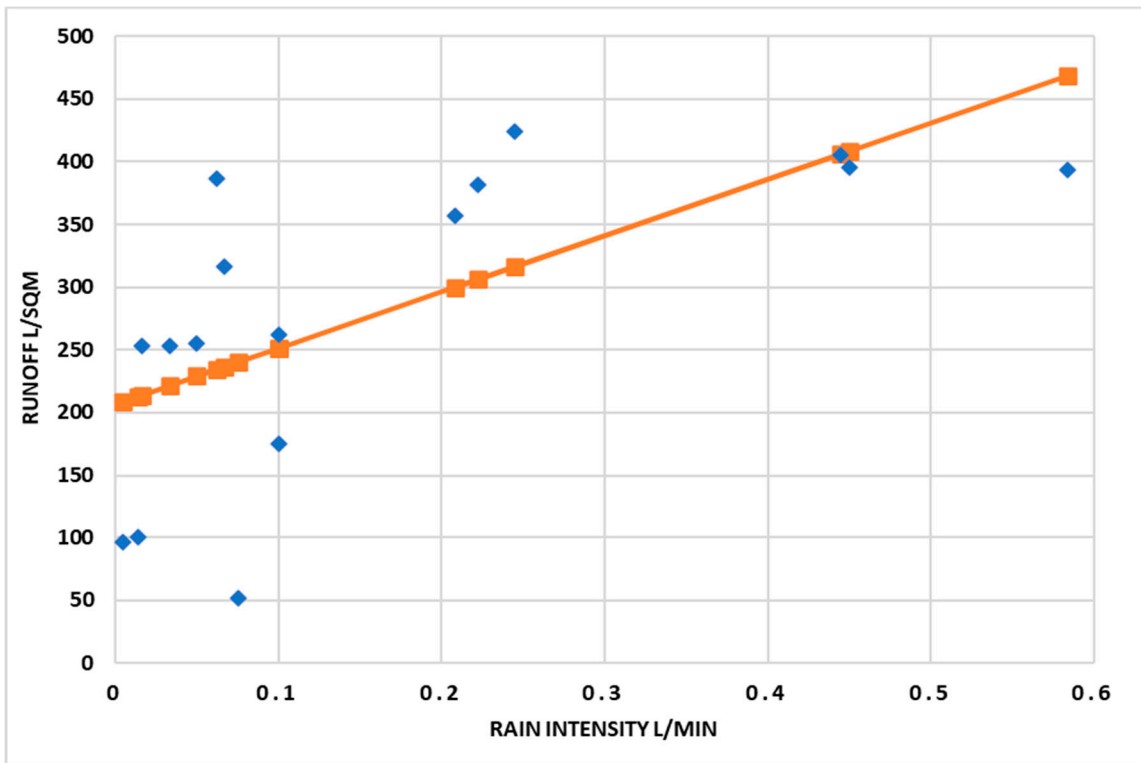

**Figure 7.** The influence of rainfall stated using linear regression on runoff in compartment 49.

Within compartment 73, the regression equation obtained is: y = 206.61 + 4380.408x (r-square = 0.38), as represented in Figure 8. According to this, the surface runoff is influenced to the extent of 38% by the intensity of the rain. Comparing the calculated "t" value 3.219 with the theoretical t 0.01 = 2.977, it follows that the obtained results are significant.

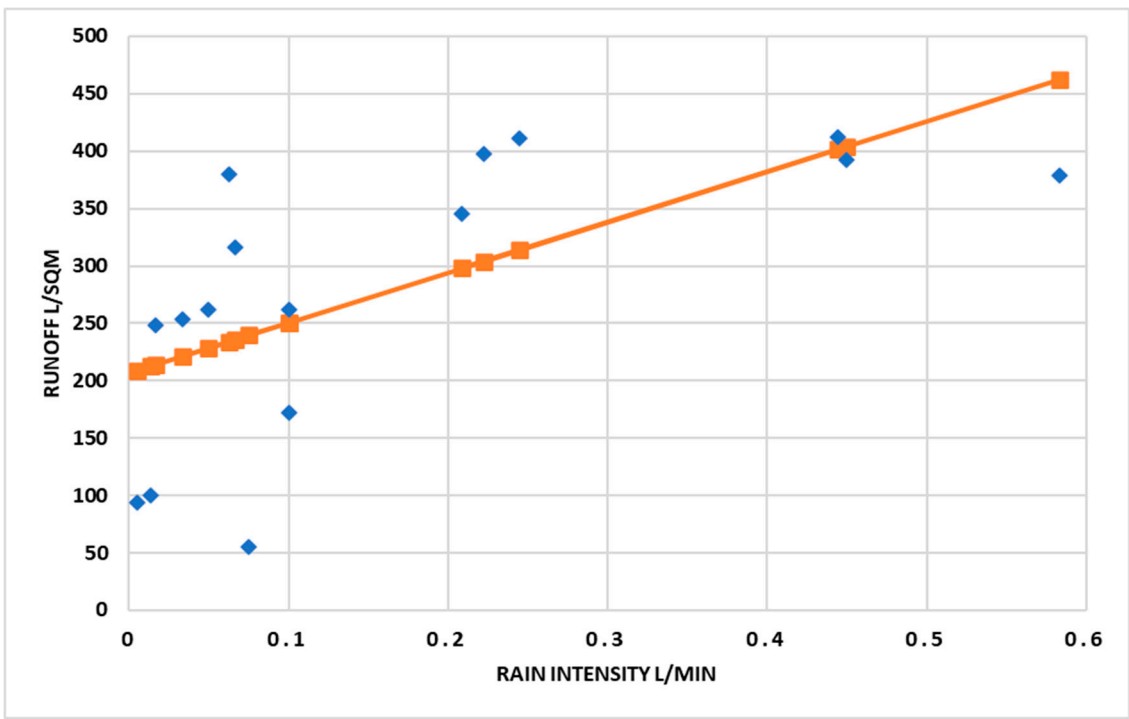

**Figure 8.** The influence of rainfall stated using linear regression on runoff in compartment 73.

As with rainfall intensity, the influence of soil retention on surface runoff was determined by applying a simple linear regression.

In compartment 49, the regression equation obtained is: $y = 192.12 + 0.0216x$ (r-square = 0.28), and its graphic presentation is in Figure 9. This means that surface runoff is influenced to a degree of 28% by ground retention. Comparing the calculated "t" value 2.558 with the theoretical t 0.05 = 2.145, it follows that the obtained results are significant.

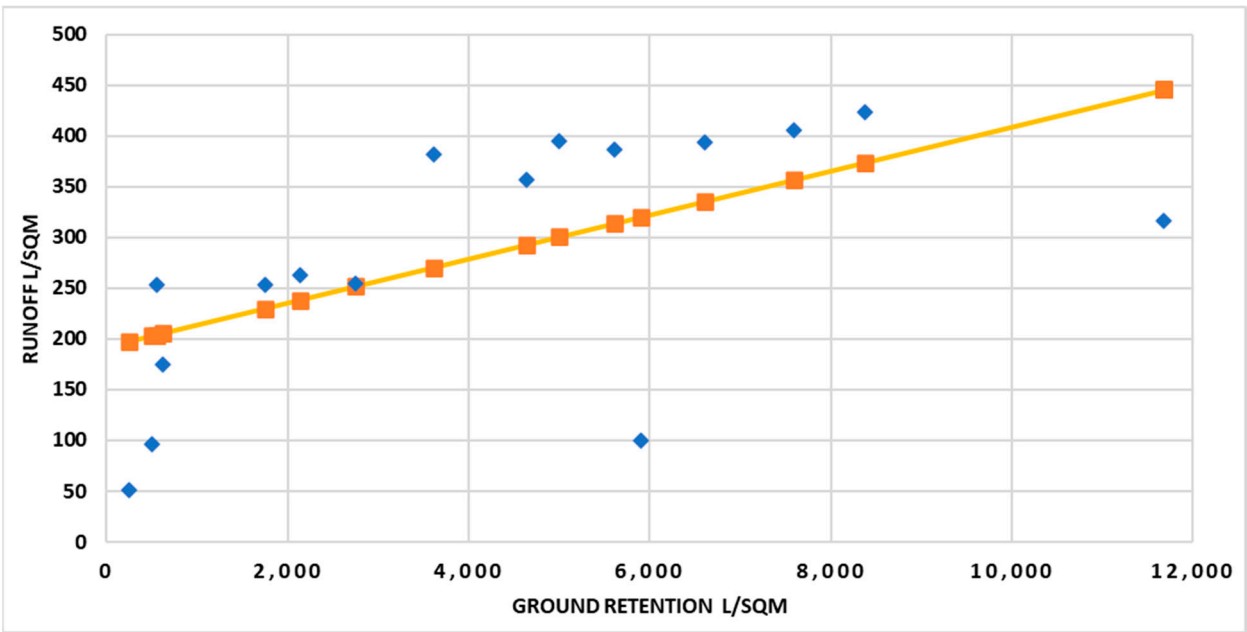

**Figure 9.** The influence of ground retention stated using linear regression on surface runoff in compartment 49.

In compartment 73, the regression equation obtained is: $y = 192.81 + 0.0211x$ (r-square = 0.28), and its graphic presentation is in Figure 10. This means that surface runoff is influenced to a degree of 28% by ground retention. Comparing the calculated "t" value 2.513 with the theoretical t 0.05 = 2.145, it follows that the obtained results are significant.

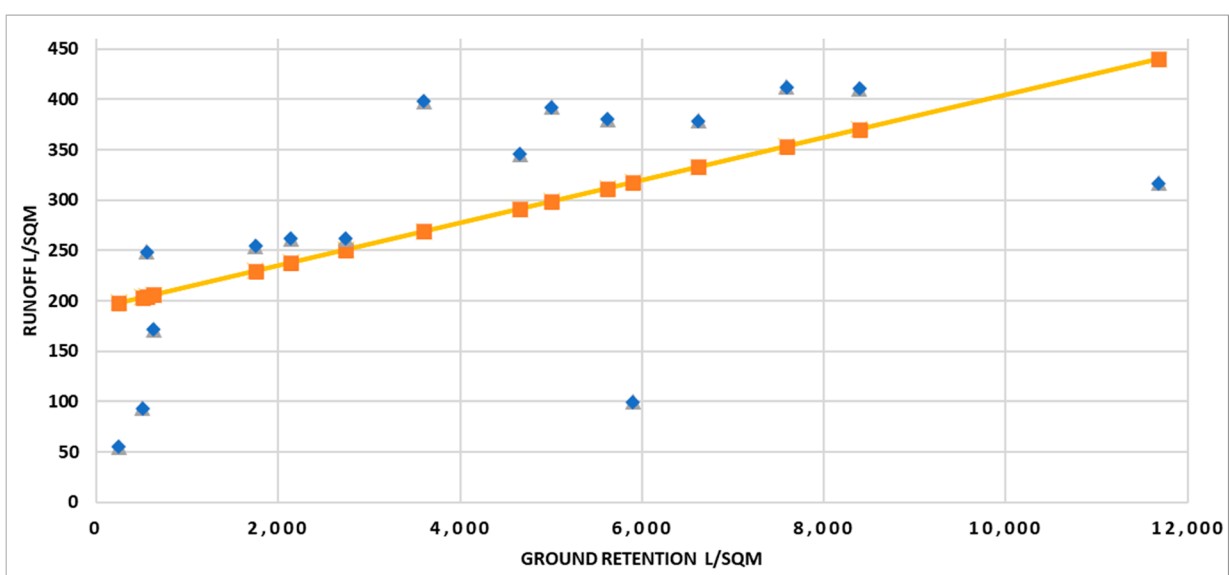

**Figure 10.** The influence of ground retention stated using linear regression on surface runoff in compartment 73.

## 4. Discussion

In the present study, the behavior of some pine stands installed on lands with erosion phenomena was examined. They, as an ecosystem, fulfill their function of protecting the soil from surface runoff through canopies, grassy vegetation, shrubs, and litter [30–32]. The more vegetation that covers the ground, the more precipitation quantity is intercepted [37,38], and as the litter is deposited on the ground, the degree of infiltration increases [39] and decreases the volume and speed of runoff [40–42]. Thus, the soil particles' detachment phenomena are generated by water drops [42,43].

In order for these aspects to be correlated, it was necessary to collect data on the amount of rainfall water that reached the ground level, represented by the rain intensity as directly influenced by the phytosanitary condition of the forests and implicitly influenced by the consistency of the stands [35]. The characteristics of the stands studied influenced water runoff and, therefore, soil erosion [44–47]; in percentage terms, the anti-erosion effect was 64% in sub-compartment 49 and 66% in compartment 73. Lukic et al. [48] showed that the water runoff on the slope was considerable, and implicitly the estimated soil losses in forested areas were drastically reduced when compared to the estimates of soil losses immediately before afforestation, estimating a reduction of 71% and 76%. In the subsequent period to date, soil loss under the black pine grove has been reduced by 58%. Durán et al. [49] reported a 58–98% reduction in soil loss in areas covered with different types of vegetation compared to areas without vegetation, while [44] suggested a decrease of up to 70–95%. They observed a significant reduction in soil losses in the first 10 years after black pine afforestation, while such a reduction has not been statistically significant since the year 11 until now.

The high short-term erosion rates obtained by Aburto et al. [9] suggest that the evaluated soils are more fragile as they displayed average sediment yields in the high ranges for world forest lands. A global analysis of erosion rates under different land coverages conveyed that, for most land uses and coverages, there is a wide range of values reported in the literature and that the ranges of values are highly dependent on the method used to estimate erosion rates. For direct measurements in forest lands, which included both plantations and native forests, erosion rates vary from approximately 0.2 to about 104 mg ha$^{-1}$ yr$^{-1}$ [5]. The higher estimate for this range corresponds to a recently harvested forest on a steep hillslope site (bare and fallow) in China, which is similar to our highest estimate. Similarly, Borrelli et al. [6], using erosion pins, estimated average erosion rates of 49.1 mg ha$^{-1}$ yr$^{-1}$ for recently-secondary native forest harvested sites in the Italian Central Apennine Region, which are similar to our erosion rates for harvested plantation sites (52.0 mg ha$^{-1}$ yr$^{-1}$).

Attempts to quantify soil erosion from runoff have been pursued for at least 4000 years, but recently the problem has become a serious one, and efforts have been made to obtain global erosion data as a prelude to soil conservation [50]. Surface runoff along a partially saturated soil can either accumulate downstream or seep into the soil. Due to the transient nature of soil and sediment flows, drainage networks are generally transient [51]. Microflows occur at a certain critical distance downstream on the slope and continue in parallel for a certain length before merging with larger seeps. Thus, surface runoff detaches soil particles that have been detached by precipitation and transports them downstream, where they settle [52].

The present study and others [53,54] show that vegetation cover associated with rainfall regulates the generation of surface runoff and thus significantly reduces soil erosion. That is why the stands installed on degraded land must be carefully monitored, applying the necessary works in time in order to maintain mechanical stability and good phytosanitary conditions. It is also recommended to apply some measures, where necessary, to promote the natural regeneration of shrubs, seedlings of arborescent species, and grasses to increase the anti-erosion capacity of stands.

Lou et al. [55] found that surface runoff has a positive correlation with increased recorded rainfall. According to the authors, this could be justified by the fact that during early rainfall stages, the soil moisture content is low while the soil is not fully saturated,

causing preliminary raindrops to fill empty soil pores without generating surface runoff. This was also found by Chen and Wang [56], who previously confirmed that the surface runoff is relatively small at the beginning of the rainfall event.

In the semi-arid environment of the Mediterranean area, they found that the effect of spatially-structured vegetation on runoff and sediment production was very dynamic and depends on the intensity of precipitation and the slope of the terrain. With the amount and intensity of rainfall above certain thresholds [57], the effect of vegetation on surface runoff and erosion can be weakened [58,59]. Nainar et al. [60] found that in plots with litter, surface runoff increased linearly with precipitation. This follows the findings of other existing studies that found similar relationships in a secondary mixed forest [58]. Zheng and Jia [61] had a modern approach to interception in crowns, determining with the help of satellites and aerial images and using the Gash model that is widely applied to different land cover types, such as rainforest, coniferous forest, broad-leaf forest, shrubs, and crops [62]. It has been proven to be able to estimate regional and global interception loss with high accuracy and efficiency when combined with a satellite-remote sensing data set [56,63].

The restoration of degraded lands involves quite high costs in Romania in 2022, starting from EUR 20,000 per hectare until reaching the massive state (transition from plantation to forest stage), depending on the type and percentage of soil degradation.

Although the financial and technical implications are quite high, and the economic gains from wood material are almost non-existent, from an ecological point of view, these lands are introduced into the eco-productive circuit, while at the same time, the climatic conditions in the neighboring areas are improved.

## 5. Conclusions

The consistency of trees, closely related to the number of trees per hectare and associated with shrubs, seedlings, and litter, has a positive effect on the interception of precipitation and thus reduces its speed and intensity in stands. This reduces the process of erosion caused by precipitation.

In order to be able to appreciate the anti-erosion efficiency of the forest, the influence of the intensity of the rains and of the soil retention on the surface runoff was followed by means of simple linear regression, through which it was shown that the two parameters influence the runoff in the same percentages. Analyzing the two parameters, it was observed that the degree of proximity to the crowns directly influenced the intensity of the rain felt within the forest, which indirectly affected the leak. In addition, the coverage of vegetation (trees, seeds, shrubs, herbaceous blankets) through the litter layer that is formed favors retention at the ground level and thus creates areas that favor infiltration by reducing water speed. In percentage terms, the quantified anti-erosion effect is 64% in compartment 49 and 66% in compartment 73. Although the results obtained were in line with the ones mentioned in the specialized literature, the study was limited by the main climatic factor: precipitation during the years in which the measurements were carried out. As the precipitation was relatively reduced, it can be mentioned as a limit of the study. Another limitation consists of the fact that the installation of artificial rain devices was impossible due to the trees and the lack of water sources.

In order for the anti-erosion efficiency of stands installed on degraded lands to be known, this type of study should be extended to various stands from various areas of the country. Furthermore, following the results, the afforestation norms can be updated or new ones created. Moreover, in order for these stands to maintain their anti-erosion efficiency, it is recommended to analyze the current risk factors present in the amelioration perimeters and create some maps by degrees of risk (climatic, pedological). When applying silvicultural operations, they should especially promote natural regeneration, especially in areas where risks of landslides or floods are present.

**Author Contributions:** Conceptualization, M.M., I.T., F.A.R. and M.D.; methodology, M.M. and M.D.; software F.A.R. and B.S.; validation, I.T., M.D. and I.D.A.; writing—original draft preparation, M.M.; writing—review and editing, I.T., M.D., F.A.R. and B.S.; visualization, I.T., M.D., F.A.R. and B.S.; supervision I.T. and M.D.; project administration, M.M., I.T. and M.D.; funding acquisition, M.M., I.T. and I.D.A. All authors have read and agreed to the published version of the manuscript.

**Funding:** This research received no external funding.

**Institutional Review Board Statement:** Not applicable.

**Informed Consent Statement:** Not applicable.

**Data Availability Statement:** Not applicable.

**Conflicts of Interest:** The authors declare no conflict of interest.

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
