# Peer review of "Determining the Anti-Erosion Efficiency of Forest Stands Installed on Degraded Land"

_sustainability, doi:10.3390/su142315727_

Round 1

Reviewer 1 Report

The manuscript has evaluated the anti-erosion effect of stands established on degraded land, which is significant for relevant policies make for better soil erosion prevention. But there still has some room for improvement. The 1-2 main concerns must be modified, and the third one is a suggestion that could improve the completeness of this paper. Detailed comments for this paper are as below.

Main concerns:

1. The main purpose of this paper is to evaluate the anti-erosion effect of stands established on degraded land. But the main analysis was conducted between leakage, rain intensity, and ground retention. What’s the relationship between them? Is there any previous study that could support this method?

2. The discussion section needs to be rewritten, more like an introduction that introduced what was found by previous studies. This section should concentrate on the main conclusion of this study.

3. As mentioned in lines 56-60, the economic incentive was a key factor for the farmer to accept the plan. Some discussion about the cost and gain for different stands could be simply introduced, which could make it more complete.

Minor comments:

1. Line 12, “the reasons of forest soils degradation” should be modified into “the reasons for forest soils degradation”.

2. Lines 19-21, the sentence “we analyzed the consistency…of rainfall and soil retention” should be rewritten.

3. Lines 24-25, the sentence “Indirectly surface runoff is…the intensity of rain” should be rewritten.

4. Line 51, the sentence is lack a “.”.

5. Table 1 should be modified, which is hard to see clearly, please adjust the width properly.

6. Line 150, the “ronoff” may be wrong.

7. Line 195, the title of Figure 4 needs to be checked.

8. Line 257 and 258, the word “com-pared” and “reduc-tion” need to be checked.

9. Line 274, the use of reference 6 should be modified as the style of the journal.

Reviewer 2 Report

[Be sure to review my suggestions made on the manuscript itself - uploaded along with this review report.]

Comment on title:

Once one begins reading the paper, it is clear that the authors are referring to forested stands, but in the title, it is unclear. They may be referring to hunting stands, display stands, food stands, etc. I recommend inserting the word “forest” in the title to improve searchability of the article for those interested in forestry and erosion. Perhaps this should also be done in the first sentence of the abstract for similar reasons.

To focus more on the scientific soundness of this paper, I stopped “readability” editing at the end of the Introduction due to how much time this takes. However, similar editing needs to be continued throughout the remainder of the manuscript.

Comments on Methods:

Manuscript implies that soil retention is estimated via precipitation-based measurements, citing Dîrja, M., Pepine, A., 2008, AmelioraÈ›ii silvice îndrumător pentru întocmirea proiectului, Publisher Todesco: Cluj-Napoca, Ro-390 mania, pp. 10-44. This paper is unknown to me, and cannot be found in any search engine I have tried.  The authors imply that their study shows the effect of precipitation (as altered by crown cover, tree density, seedling species) on soil retention, but I don’t think they can make any claims here, especially since they don’t measure total solids in the water samples. The equation they provide in the paragraph that describes the methods used for estimating soil retention predicts surface runoff, not solids concentration. While there are many studies that measure surface runoff and solids concentration and show that there is a correlation under many conditions, this paper, I believe, cannot claim any amount of erosive effect based on precipitation/runoff measurements they present. They should limit their manuscript to rainfall/runoff discussions, and in the conclusions they might allude to the many studies that show there is a positive correlation between precipitation and soil retention when healthy forested stands exist.

Cannot read the legend in Figure 1.

Comments on Results:

The authors report statistical results in an unconventional manner, apparently using asterisks to indicate level of significance rather than simply reporting r-squared and p values. Also, it is far more common when using multi-regression techniques to make statements such as “independent variable Q explains 39% of the variation of dependent variable G”. So, for example, in this paper the statement beginning on line 211 would more typically be written something like: Within compartment 49, the regression equation obtained was: y=206.73+448.19x (r-squared=0.73, p=0.01), with rainfall intensity explaining 39% of the variation in leakage runoff.

When referring to rainfall intensity, typically we are referring to amount of precipitation per unit time (in fact on line 169 the authors show this). However, in the results the authors seem to be using amount per unit area (but I am not familiar with units of l/mp). Be sure the correct units are being reported in the manuscript, and ideally use an uppercase L for liters.

Round 2

Reviewer 1 Report

The manuscript has been modified based on the comments of the first round review, but there still has some problems that need to be revised. The relevant reference that explained the relation between surface flow and soil erosion needs to be added to evidence the relevant description. A discussion about the linkage between surface flow and soil erosion also needs to be added to connect the results of this study and soil erosion. Finally, the writing needs to be checked carefully, some words are still wrong in the revised manuscript.

Reviewer 2 Report

I would encourage to review my comments on the pdf file and see if the authors addressed any of them

Author Response

Dear reviewer, we have taken into account all the changes proposed in the pdf version. Please see the latest version of the article. Thank you for everything. 

Round 3

Reviewer 1 Report

The manuscript has been modified based on the comments of the previous review, most of the suggestions were settled but still have some spelling problems in the manuscript. I suggest accepting the manuscript for publication after these problems were solved.

1. Line 34, “sinificant” should be changed into “significant”.

2. Figure 3 needs to be redrawn.

3. Line 262, “dtacment” may be changed into “detachment”.

Reviewer 2 Report

The authors have sufficiently addressed my concerns in the revised manuscript. There are still some English language issues that should probably be corrected prior to publication (e.g. many examples of 1-sentence paragraphs, the use of the word "respectively" has been misused in the context of the sentence, etc.)
